# Effect of Air-Polishing and Different Post-Polishing Methods on Surface Roughness of Nanofill and Microhybrid Resin Composites

**DOI:** 10.3390/polym14091643

**Published:** 2022-04-19

**Authors:** Kinga Dorottya Németh, Dóra Haluszka, László Seress, Bálint Viktor Lovász, József Szalma, Edina Lempel

**Affiliations:** 1Department of Restorative Dentistry and Periodontology, University of Pécs Medical School, Dischka Gy. Street 5, 7621 Pécs, Hungary; kinga.nemeth3@gmail.com; 2Department of Biophysics and Radiation Biology, Semmelweis University, Tűzoltó Street 37–47, 1094 Budapest, Hungary; haluszka.dora@med.semmelweis-univ.hu; 3Central Electron Microscope Laboratory, University of Pécs Medical School, Szigeti Street 12, 7624 Pécs, Hungary; laszlo.seress@aok.pte.hu; 4Department of Oral and Maxillofacial Surgery, University of Pécs Medical School, Dischka Gy. Street 5, 7621 Pécs, Hungary; balint10@hotmail.co.uk (B.V.L.); szalma.jozsef@pte.hu (J.S.)

**Keywords:** air-abrasion, enamel, microhybrid, nanofill, polishing, resin-composite, surface roughness

## Abstract

Air-abrasion is a popular prophylactic procedure to maintain oral hygiene. However, depending on the applied air-abrasive powder, it can damage the surface of the tooth and restorations, making it susceptible to plaque accumulation. The purpose of this study was to investigate the effect of 5 s and 10 s air-abrasion of calcium carbonate on surface roughness (Ra) of enamel, nanofill, and microhybrid resin-composites and the effect of post-polishing with two-step rubber- (RP) or one-step brush polisher (BP) to re-establish the surface smoothness. Surface topography was visualized by scanning-electron-microscopy. The quantitative measurement of the Ra was carried out with atomic-force-microscopy. Air-abrasion for 10 s decreased the Ra of enamel as a result of abrasion of the natural surface texture. Post-polishing with RP after 10 s air-abrasion did not change the Ra or BP; however, Ra was increased significantly by scratching the surface. Air-abrasion increased the Ra of resin composites significantly, irrespective of the application time. While RP provided a similarly smooth surface to the control in the case of microhybrid resin composite, BP increased the Ra significantly. The Ra for the control group of the nanofill-resin composite was initially high, which was further increased by air-abrasion. RP and BP re-established the initial Ra with deeper and shallower scratches after BP. Both the material and treatment type showed a large effect on Ra.

## 1. Introduction

Resin-based composites (RBC) are one of the most common types of restorative materials in dental clinical practice. A great volume of research is currently ongoing in an effort to maximise the degree of conversion, color stability, abrasion resistance, and polishability, as well as to minimize polymerization shrinkage.

The surface structure of an RBC filling is very different from that of tooth enamel, exposed dentin, or cementum. The properties of the filling material under oral conditions depend, among others, on the composition of the material, the degree of monomer to polymer conversion, the polishing process, and the application of prophylactic treatments [1]. Polishing to the right extent is an essential step in making a long-lasting RBC restoration. This working phase influences the surface morphology, hardness, roughness, and surface gloss of the RBC restoration, and thus its aesthetic appearance and abrasion resistance [2]. RBCs are expected to have adequate surface properties while avoiding undesirable biological interactions in the oral cavity, including microbial adhesion and biofilm formation. The presence of dental plaque is the main cause of secondary caries and development of periodontal lesions [3,4]; therefore, individual and professional cleaning is mandatory in caries prevention.

In addition to the traditional removal of dental calculus, professional tooth cleaning with brushes, rubber cups, and pastes, the use of various prophylactic abrasive powders with water, namely air-polishing (AP), is a common way for removing dental biofilm and discoloration. AP can be considered an alternative to conventional techniques. It is a highly effective, easy, and rapid technique which gives rise to less operator fatigue with improved access to hardly accessible tooth surfaces [5,6]. Several air-polishing systems are available in the dental practice to remove deposits from the supra- and subgingival tooth surfaces. The application field of such abrasives also includes cleaning orthodontic braces, arches, implant surfaces [7,8]. Among the prophylactic abrasive powders, sodium-bicarbonate was the first to be marketed, with a particle size of up to 250 µm. This conventional type is highly abrasive and destructive, leaving damage and roughness on the surface of the restorations as well as on the enamel [6]. With the intention to eliminate the negative effects, glycine, calcium carbonate, reduced particle size sodium-bicarbonate, and erythrol-based prophylactic powders were introduced in the early 2000s [9,10]. During their use, the particles reach the tooth surface in a controlled radius accompanied by air and water. The new types of abrasive powders however can also cause material loss on the surface of the enamel, dentin, cementum, or restoration, causing permanent changes in the surface structure [11].

Due to the relatively short history of these AP powders, there is generally a lack of information about their exact effect on the surface properties of enamel and RBCs. Rough surfaces tend to be a substrate for stain accumulation as well as dental biofilm adhesion [12]. Such biofilm then may damage the mineralized tissues or contribute to infections of the soft tissues. Furthermore, bacteria invading the interface between the tooth and the restorative material are the principal etiologic factors responsible for secondary caries [13]. Although several studies have already showed the negative effect of AP, no recommendation describes the necessity of restoring the structural changes of enamel or RBCs, which would provide an optimal polish that favors the aesthetic expectations of the patient and improves dental and periodontal health.

One way to objectively measure the surface irregularities in a 3D perspective is with the application of an atomic force microscope (AFM). AFM is a powerful and widely-used technique for detecting local structural and mechanical properties of various surfaces with high resolution. In the AFM, a cantilever with a tip scans the specimen surface, and the topological features are mapped with nanometer resolution. This technique is based on the detection of attraction and repulsion interactions between atoms of the surface to be examined and the nano-sized cone shaped tip integrated on a cantilever.

The purpose of our in vitro study was to test the effect of the calcium carbonate-based prophylactic AP method on nanofill, microhybrid RBCs as well as enamel and to obtain a direct quantitative comparison of the changes in surface roughness using an AFM. Our further aim was to investigate the efficiency of one-step and two-step post-air-abrasive polishing processes on surface roughness to re-establish the baseline surface smoothness. The null-hypotheses of our investigation were the following: (a) there is no difference in the surface roughness of resin composites and native enamel surfaces before and after the air-polishing with calcium carbonate; (b) there is no difference in the surface roughness among the resin composites with different filler compositions after air-polishing with calcium carbonate; (c) there is no difference in the surface roughness among the resin composites and on the enamel after different post-polishing procedures.

## 2. Materials and Methods

### 2.1. Sample Preparation

In this study, 8 × 5 mm enamel slices, 2 mm in thickness, were prepared from the buccal surface of unerupted third molars extracted for orthodontic reasons. The third molars were immersed to 5.25% sodium-hypochlorite solution for 5 min immediately after the extraction for disinfection, then were stored in 0.9% sterile physiologic saline solution at 37 °C (Cultura Incubator, Ivoclar Vivadent, Schaan, Liechtenstein) for one week right before the slice preparation to avoid desiccation. Ethical approval (No. PTE/3795) was obtained from the Regional Research Ethical Committee of University of Pécs. Light-cured RBC samples were fabricated using a cylindrical polytetrafluoroethylene (PTFE) mold (with an inner diameter of 6 mm and height of 2 mm) from nanofill (Filtek Ultimate) and microhybrid (Enamel Plus HRi) RBCs. Materials were handled as specified by the manufacturer (Table 1).

The specimens were irradiated through a polyester strip (Mylar, Dentamerica Inc., San Jose Ave, CA, USA) with a Light Emitting Diode (LED) curing unit (LED.D, Woodpecker, Guilin, China; λ = 420–480 nm; 8 mm exit diameter fiberglass light guide) in standard mode for 20 s according to the manufacturer’s instruction, powered by a line cord at room temperature. The irradiance of the LED unit was 1050 ± 10 mW/cm^2^ and monitored before and after curing with a radiometer (Cure Rite, Dentsply, Milford, DE, USA). All enamel and RBC samples were stored in distilled water at 37 °C for maximum one week prior to testing.

The prepared samples were randomly divided into five groups (n = 5 × 5). Surface characteristics of materials cured against a Mylar strip were used as control (Group 1). Samples in Group 2 were air-polished (Prophy-Mate Neo, NSK-Nakanishi Co., Kanuma, Tochigi, Japan) with a 54-µm particle size calcium carbonate prophylactic powder (Mohs Hardness Index: 3) (Prophy-Mate Profilactic Powder, NSK-Nakanishi Co., Kanuma, Tochigi, Japan) for 5 s at a 20-degree angle from a distance of 5 mm. Samples in Group 3 were air-polished for 10 s with the same parameters. The powder chamber was refilled after each AP series, to ensure the maximum reproducibility of the device. Samples in Group 4 and Group 5 were post-polished after the AP. In Group 4, a two-step rubber diamond polisher (fine—10 s, 8–32 µm grit size, Kenda Nobilis, Kenda AG, Vaduz, Liechtenstein; then extra fine—10 s, 4–8 µm grit size, Kenda Unicus, Kenda AG, Vaduz, Liechtenstein) was used. In Group 5, a one-step polisher, an abrasive-impregnated polishing brush with built-in silicon carbide abrasive particles (Occlubrush cup, KerrHawe SA, Bioggio, Switzerland), was used for 10 s to polish the surface (Table 2).

### 2.2. Surface Morphology Analysis

Samples were analyzed by scanning electron microscopy (SEM) imaging as a preliminary test. The SEM examination was used to evaluate the clinical significance of the effects of AP on each type of material treated in the study. For this process, each enamel slice and RBC sample underwent an acetone dehydration series and then was sputter-coated with a golden layer (fine coat Ion sputter JFC-1100). The pretreated samples were examined under 1000× and 2000× magnification with a JSM-6300 (JEOL, Tokyo, Japan) type scanning electron microscope. To analyze the SEM images, the effect of prophylactic AP and post-polishing processes on each enamel and composite surfaces were examined. All destructive surface changes compared to native enamel and RBC surfaces were considered clinically significant.

### 2.3. 3D Surface Topography Analysis

The samples were analysed with AFM to provide information about the exact surface topography as well as changes in the mean height values of irregularities. This process is based on the detection of attraction and repulsion interactions between the atoms of the surface to be examined and the nanoscaled cone-shaped tip integrated on a cantilever. Samples were fixed to the probe holder using a liquid adhesive (Loctite 406, Henkel, Düsseldorf, Germany) which was then attached to a slide adapter. Surface roughness was determined using an AFM unit (Asylium Research, Santa Barbara, CA, USA) synchronized with an Olympus epifluorescence microscope (Olympus, Tokyo, Japan). In this measuring arrangement, the position of the cantilever and the sample can be set simultaneously using a mechanical stage. First, to provide a position-feedback, the cantilever was positioned next to the sample, with the laser beam aimed onto it. Then, the resonance frequency (~300 kHz) of the silicon tip (OTESPA R3, Bruker, Camarillo, CA, USA) was tuned. Before scanning, the cantilever was lowered onto the sample, so that this point could be considered as the reference or 0 point. Due to the unevenness of the sample surfaces, points higher and lower (“hills and valleys”) in relation to this point, i.e., both positive and negative values, were obtained. In order to determine the surface roughness, the smallest value of the obtained range, i.e., the deepest point, was corrected to zero. Average height values were calculated with the application of a Gaussian curve. Images were taken in a non-contact mode, with a line scan frequency of 0.3 Hz. For each sample, 30 μm × 30 μm images were scanned in three randomly selected areas at a resolution of 512 × 512 pixels. The average surface roughness (Ra) and 3-dimensional images were obtained and analyzed with the designated AFM software (IgorPro 6, WaveMetrics Inc., Lake Oswego, OR, USA).

### 2.4. Evaluation and Statistical Analysis

The altitude contrast images were analysed using algorithms built into the AFM control software. After data correction, the mean height of each sample was determined. 

Pilot study results and sample size formula were used to estimate sample size.

Sample size formula:$$n=\frac{{\left(z_{1-\frac{\alpha }{2}}+z_{1-\beta }\right)}^{2}{\left(s_{1}+s_{2}\right)}^{2}}{{\left(M_{1}‒M_{2}\right)}^{2}}=2.5$$
(*z* = standard score; α = probability of Type I error = 0.05; *z*_1 − α/2_ = 1.96; *β* = probability of Type II error = 0.20; 1 − *β* = the power of the test = 0.80; *z*_1 − *β*_ = 0.84, *M*_1_ = 1.71, *s*_1_ = 0.14, *M*_2_ = 1.31, *s*_2_ = 0.18). By adopting an alpha (α) level of 0.05 and a beta (*β*) level of 0.20 (power = 80%), the predicted sample size (*n*) was found to be a total of 2.5 samples per group. Instead of the calculated 2.5 samples, *n* = 5 per group sample size was selected.

One-way analysis of variance (ANOVA) and Tukey’s post-hoc test was used to compare the average height of the untreated sample of a given group and the average height of the different surface-treated samples. Relative effect sizes for factors *Material* and *Treatment* as well as their interactions on the surface roughness of the investigated resin-based composites were analyzed by General Linear Model and Partial Eta-Squared statistics. The difference was accepted as significant at the 95% confidence level where *p* < 0.05.

## 3. Results

The SEM evaluation provided a visual qualitative analysis of changes on the enamel and RBCs’ surfaces as a result of AP treatment (Figure 1). As a comparison, in the case of enamel samples, at magnification 1000×, the images of the control group seemed to show the enamel prisms to be well structured and more visible. In the AP or post-polished groups, these surface characteristics were less dominant. The surfaces of the AP-treated RBC samples exhibited strong surface roughness compared to both the control and post-polished specimens, although, the control group of FU RBC already showed a slightly rougher morphology. A magnification of 2000× enabled the detection of differences in surface structure between the samples air polished for 5 s and 10 s. Longer exposure to air-polishing resulted in more intensive surface roughness. Disintegrated matrix and free filler particles were visible both on the degraded nanofill and microhybrid RBC surfaces.

As well as the volumetric loss, the incorporation of the Ca-carbonate profilactic powder into the RBC surface may account for the more characteristic changes. With the rubber post-polishing method, the surface roughness seemed to be restored; however, polishing with a silicon carbide brush was less effective and caused defects with a different surface character.

The results of the AFM study allowed the changes in the surface structure to be quantitatively analysed. The software produced a frequency distribution of the height value assigned to each pixel of the scanned area. Three-dimensional measurements were used to examine the mean height differences between each sample and group.

As measured by AFM, while 5 s air-polishing did not influence the enamel surface roughness significantly, air-polishing applied for 10 s decreased the natural microtexture of the enamel by a significant degree. Rubber polishing after 10 s of AP did not change the surface characteristics; however, polishing with a brush containing silicon carbide particles increased the roughness significantly, creating a similar mean height of irregularities as was detected initially, although with different surface character (Figure 2 and Figure 3, Table 3).

The SEM and 3D AFM images of the FU nanofill control group showed an initially rough surface with a high frequency of hills and valleys. The surface roughness value obtained for the FU_5s_AP sample differed significantly from the control with a decrease in the density of hills and valleys. The FU_10s_AP group showed a slightly rougher surface; however, this was not significant. Post-polishing methods achieved significantly smoother surfaces which was similar in extent to the air-polished (Figure 1, Figure 3 and Figure 4, Table 4).

For the control group of EP, microhybrid RBC, SEM, and AFM analysis showed a relatively smooth surface morphology which was significantly roughened both by the 5 s and 10 s air-polishing. Rubber-polishing provided an even surface with similar mean surface roughness values compared to the control group. Polishing with a silicon carbide brush could not decrease the roughness of the air-polished samples (Figure 1, Figure 3 and Figure 4, Table 5).

A 2 (*Material*) × 2 (*Treatment*) mixed-model ANOVA revealed that the main effect for both the RBC *Material* and *Treatment* on surface roughness values was significant with a large Partial Eta-squared value. The interaction (*Material × Treatment*) of the two factors also showed a large effect on the Ra values (Table 6).

## 4. Discussion

This in vitro study analysed the effects of a prophylactic AP and two different post-polishing methods on the surface roughness of a nanofill and a microhybrid RBC as well as on a native enamel surface. Our results showed that the surface characteristics were material dependent and also vary in relation to each surface treatment method. Thus, all three null-hypotheses are rejected. The significance of the surface roughness of the tooth structure and dental restorative materials lies in the fact that the biofilm accumulation increases as the Ra increases [14,15] and reduces significantly below an Ra of 0.2 µm [4]. Plaque accumulation subsequently increases the risk of caries, gingival irritation, and periodontal inflammation [16]. Additionally, Ra may lead to staining, which compromises the aesthetic appearance [17].

AP is defined as the reduction of deposits or surfaces by abrasive particles suspended within a moving fluid [7]. Although AP is thought to be a minimally invasive stain removing method which does not damage sound tooth tissue, not all available AP powders are gentle on the enamel, dentin, and cementum [18]. Several studies demonstrated a reduction of enamel roughness. Although this would be beneficial after scaling [19,20], surface reduction may also increase susceptibility to erosive effects which can further increase enamel surface loss and may induce or intensify dentin hypersensitivity [21]. Our findings also revealed a loss of tooth structure with disappearance of the natural texture of enamel after the use of AP with calcium carbonate. This was shown by a decrease in Ra values. Extended AP time further decreased the Ra, assuming even more enamel removal. However, since roughness and not substance loss has been assessed in the present study, inference on the abrasiveness of the assessed surface treatments is not possible.

The use of rubber polishers resulted in smoother surfaces; however, this was not significant compared to a 10 s treatment with AP. As demonstrated by the SEM and AFM analysis, polishing the enamel surface with a brush containing built-in silicon carbide particles after AP is not recommended, since it can increase the Ra to a high value by creating scratches and thus making the surface retentive to plaque accumulation. Occlubrush is a one-step polishing system for all types of RBCs, compomers, resin-modified glass ionomers, and ceramic indirect restorations. According to the manufacturer’s description, the special fibers within the bristles ensure Occlubrush is non-destructive to tooth structure or to the margins of the restoration, since it provides 0.25 µm Ra on the polished surface. An earlier report described that the silicon-carbide-impregnated bristle polishing brush maintained a surface texture similar to smoothing between a 25-µm finishing diamond and an extra-fine coated abrasive disc and was not deleterious to enamel [22].

AP of restorative materials generally results in substance reduction. The amount of substance loss depends on the type of the abrasive powder applied as well as the distance, angulation, and time of the exposure [23,24]. According to the present results, the 3D effect of AP and post-polishing was dependent on the RBC type. There was a significant initial discrepancy already between the control groups of the nanofill and microhybrid RBCs. To ensure a smooth initial surface for the control groups of RBCs, the materials were cured against a polyester matrix [25]. Results of our study revealed high values of Ra in the case of untreated (control) nanofill RBC, while microhybrid RBC demonstrated a relatively smooth surface after polymerization through the polyester strip. This result could be explained by the filler and cluster particle size which is exposed on the surface. Although Filtek Ultimate contains nanoparticles between 4–20 nm, the aggregated nanoparticles form a relatively big cluster with an average particle size of 0.6–10 µm. In comparison, the particle size of the microhybrid Enamel Plus HRi is between 20 nm and 3 µm. The literature reports that the main intrinsic factor that affects the surface smoothness of the RBC is the particulate filler type, size, and quantity in the resin matrix [26]. The advantage of a nanocluster prevails during polishing and function, because abrasive effect results in only drift off of the nanomers from the cluster surface instead of the turn-out of the whole cluster. In line with the previous phenomenon, Moda et al. also found nanofill RBC to have the lowest Ra compared to microfill and microhybrid RBCs after finishing and polishing [27]. Extrinsic factors can also influence Ra. This may include abrasive and erosive effects during function and oral hygiene, thermal changes, and hydrolytic or enzymatic degradation [28]. Although better polishability and esthetic appearance of nanofill RBCs is thought to be superior to nano- or microhybrid RBCs, due to the greater surface area to volume ratio of its filler particle system, the nanofill RBC may also suffer a higher degree of degradation during functioning in the oral cavity [29]. This is confirmed by an in vivo study, which found nanofill RBC fillings to show a significantly greater color change after seven years’ service [30].

AP resulted in significantly increased Ra in both types of RBCs compared to the control groups. The effect of AP on microhybrid RBC was more than three-fold, while it was less aggressive on the nanofill RBC. The extended AP time had no significant effect on the Ra. The result may be attributed to the discrepancy among the RBCs in filler and matrix hardness and ratio. The composition, especially the filler content and particle size, as well as the ability of the polishing system to abrade the matrix and the filler, may also contribute to the observed changes in surface characteristics [31]. Pelka et al. supposed that the resistance to air-powder abrasion might be due to the quality of the interfacial bonding between the fillers and the matrix or to an improved wear resistance of the matrix itself, owing to its high elastic compliance [32].

The calcium carbonate particles of 54-µm size used for the AP in this study were described as having the advantage of a rounded spherical shape created by an agglomeration of crystals. It can be contrasted with the conventional irregular crystal shape of sodium bicarbonate and aluminium oxide particles. The manufacturer recommends to use the calcium carbonate powder supragingivally and avoid the direct spray onto cementum, decalcified enamel, and margins of the restorations [33]. The hereby observed destructive effect of calcium carbonate on enamel and RBCs is not only the result of our study but also the conclusion of several publications [18,24,32]. Although the Mohs’ hardness of the calcium carbonate (3) is less than that of RBCs (~5–7), the particle size of 54 µm can wear the surface by a mechanical process. However, more clinical studies are needed to determine the effectiveness and abrasive potential of calcium carbonate, as it was concluded by Pelka et al. [34]. From another point of view, SEM analysis helped to establish that the surface roughness was not only caused by the abrasive effect of the AP powder particles but was also due to these particles embedding into the surface structure of each RBC sample. This phenomenon can cause a further change in the surface unit. This observation is not highlighted in the literature; therefore, it needs more follow-up studies to see clearly the exact effect of these embedded particles in the long-term survival of a filling.

Post-polishing of the RBCs showed different effects depending on the type of restorative material and the polishing system. In case of nanofill RBC, both polishing procedures provided similar Ra to the control group and there was no significant difference between the rubber polisher and the Occlubrush according to the topographic measurements obtained by AFM. In comparison to the rubber polished nanofill RBC, the two-step rubber polishing of microhybrid RBCs resulted in a significantly smoother surface, similar to its control group which showed originally lower Ra values. Although nanofill particles were thought to obtain better and durable polishability [35], in a systematic review carried out by Kaizera et al., it was concluded that the polishing and brightness of nanofill RBCs did not differ in a significant manner from nano- or microhybrid RBCs [36]. On the other hand, SEM images showed strongly scratched surfaces associated with higher Ra values on both RBCs after the use of the silicon-carbide-impregnated brush polisher. Silicon carbide is a hard material with a Mohs’ hardness (9–10) right below that of diamond (10), while RBCs’ hardness is between 5–7 depending on the type of RBC [37]. To provide a smooth surface, the abrasive particle of the polisher should have a higher hardness relative to the RBC’s filler particles; otherwise, the polishing agent would only remove the soft composite resin matrix, leaving filler particles protruding from the surface [38]. Besides the hardness of the abrasive particle, its size also plays an important role in the polishing efficiency. Smoother surfaces can be achieved by systematically decreasing particle size. Likewise, the polishing particle should be smaller than the RBC’s particle size to produce better results [39]. In a recent systematic review, it was concluded that the multistep polishing systems are the most effective [39]. The silicon-carbide-impregnated brush is a one-step polisher, while the rubber polisher with diamond powder applied in this study is a two-step polishing system. The use of fine (8–32 µm) followed by extra fine (4–8 µm) grit sizes could achieve a smoother surface, but only on the microhybrid RBC samples. The smoothness almost reached the bacterial attachment threshold of 0.2 µm. In contrast, the one-step polishing systems mostly achieved results further from the 0.2 µm threshold [40,41]. The polishing time for each post-polishing step was 10 s. Increasing polishing time can improve the smoothness of the RBC within a certain limitation [42]. Due to its filler content, it is supposed that better polishing results could be achieved on the nanofill RBC samples with increased polishing time.

Further investigation should be performed to overcome the present study’s limitations.

Firstly, involving more RBC- and air-abrasion types would allow for a more detailed and nuanced picture regarding the effect of air-abrasion on RBC’s surface texture. Furthermore, involving more polishing systems with different polishing times would provide the best post-polishing combination for each air-abraded RBC.

Secondly, the specimen surfaces in the present study were flat, whereas clinically, composite resin restorations have an irregular shape, which may influence the effect of both air-polishing and post-polishing. Moreover, the air-abrasion parameters (i.e., pressure, time, angle, and nozzle distance from the RBC surface) may also influence the possible effects on RBCs.

In future studies, the above-mentioned considerations should be involved to provide more detailed information about the surface characteristics of air-abraded RBCs as well as the re-establishment of their integrity with post-polishing.

## 5. Conclusions

Within the limitations of the present in vitro study, the following conclusions can be stated:(1)Air-polishing with calcium carbonate powder can cause abrasions on the enamel surface and can increase the surface roughness of both nanofill and microhybrid RBCs.(2)The destructive effect of the extended air-polishing time (10 s) is more significant on the enamel compared to the 5 s air-polishing; however, this did not influence the surface roughness of the resin-based composites.(3)The effect size of factors *Material* (type of resin-based composites) and *Treatment* (air-polishing and post-polishing) on the surface roughness is large.(4)Post-polishing with rubber polisher series can decrease the surface roughness of resin-based composites after air-polishing in a significant manner; thus, the post-polishing of resin-based composites with rubber polisher series is recommended after air-polishing.(5)Post-polishing with a series of rubber polishers has no beneficial effect on enamel which has been air-polished; however, polishing with brushes containing silicon carbide particles increased the surface roughness significantly.

As a clinical relevance, it can be concluded that air-abrasion can compromise the surface smoothness of enamel and RBC restorations; however, post-polishing with two-step rubber polishers can re-establish the surface smoothness, which may avoid increased plaque accumulation and discoloration.

## Figures and Tables

**Figure 1 polymers-14-01643-f001:**
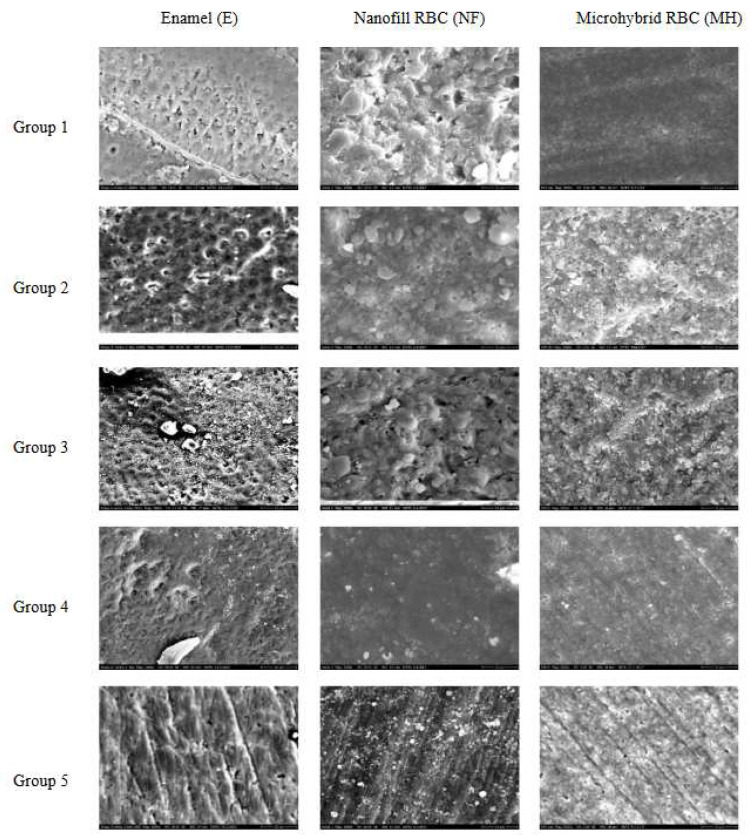
Representative scanning electron microscopy images of the control (Group 1), 5 s (Group 2), and 10 s (Group 3) air-polished and post-polished (rubber polisher, Group 4; brush polisher, Group 5) enamel (1000× magnification), nanofill, and microhybrid resin composites (2000× magnification).

**Figure 2 polymers-14-01643-f002:**
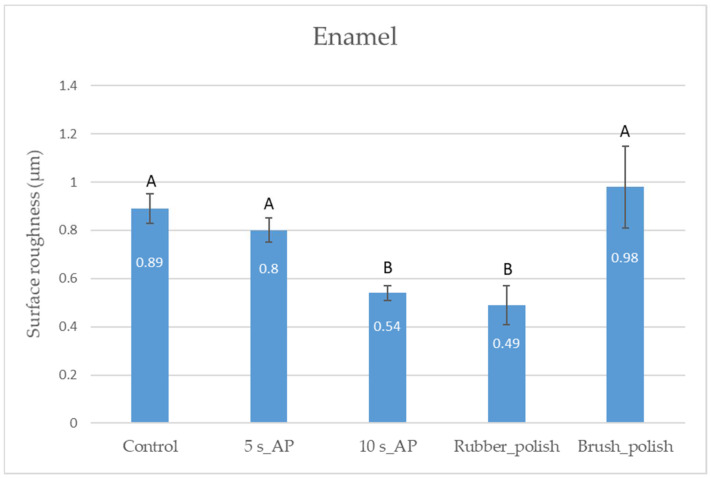
Mean values of the surface roughness on enamel measured by atomic force microscopy. Distinct capital letters (A and B) show a statistically significant difference analyzed by one-way analysis of variance and Tukey’s post-hoc test.

**Figure 3 polymers-14-01643-f003:**
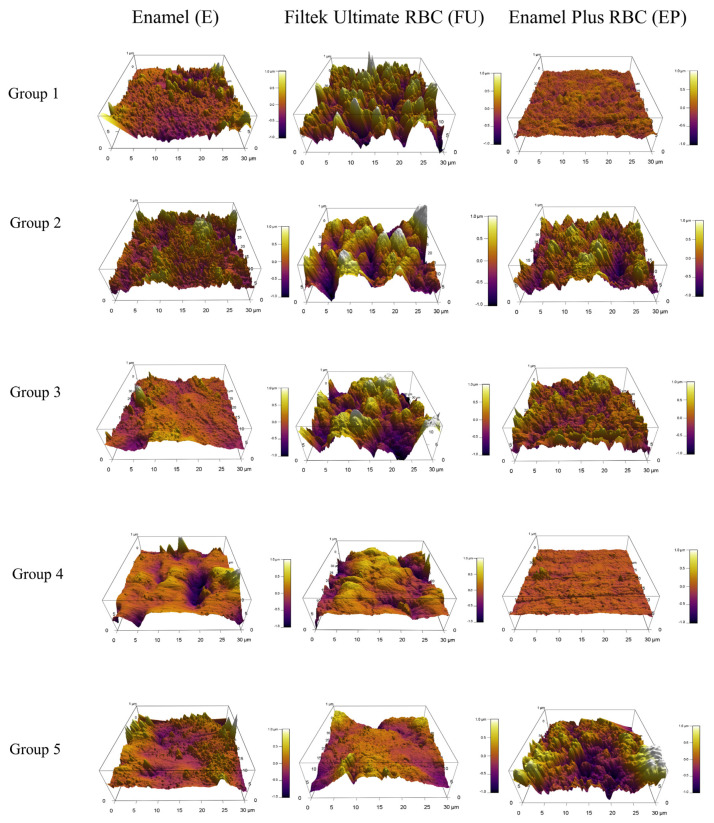
Representative 3D images of the control (Group 1), 5 s (Group 2), and 10 s (Group 3) air-polished and post-polished (rubber polisher, Group 4; brush polisher, Group 5) enamel, nanofill, and microhybrid resin composites.

**Figure 4 polymers-14-01643-f004:**
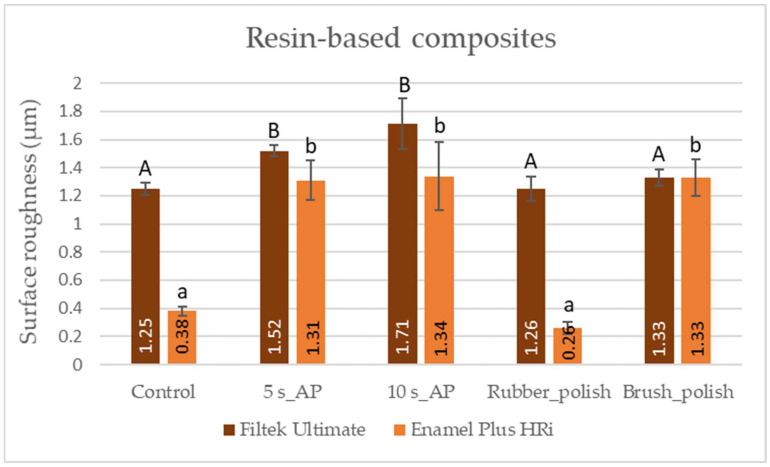
Mean values of the surface roughness on the nanofill (Filtek Ultimate) and microhybrid (Enamel Plus HRi) resin-based composite samples measured by atomic force microscopy. Distinct letters (A, B, a, b) show statistically significant differences analyzed by one-way analysis of variance and Tukey’s post-hoc test.

**Table 1 polymers-14-01643-t001:** Manufacturer, type, and composition of the investigated materials.

Material	Manufacturer	Type	Resin System	Filler	Filler Loading
Filtek Ultimate	3M ESPE, St. Paul, MN, USA	Nanofill	Bis-GMA, UDMA, TEGDMA, Bis-EMA	non-agglomerated/non-aggregated 20 nm silica filler, non-agglomerated/ non-aggregated 4 to 11 nm zirconia filler, aggregated Zr/silica cluster filler; average cluster particle size 0.6–10 µm	72.5 wt% 55.6 vol%
Enamel Plus HRi	Micerium S.p.A. Avegno, Italy	Microhybrid	UDMA, Bis-GMA, 1,4-butandiol dimethacrylate	glass filler mean size 1.0 µm, nano ZrO_2_ particles 20 nm; average filler size 0.04–3 µm	80 wt% 63 vol%

Abbreviations: wt, weight; vol, volume; BisGMA, bisphenol-A diglycidil ether dimethacrylate; UDMA, urethane dimethacrylate; TEGDMA, triethylene glycol dimethacrylate; BisEMA, bisphenol-A polyethylene glycol diether dimethacrylate.

**Table 2 polymers-14-01643-t002:** Surface treatment procedures of the investigated groups.

Group 1	Group 2	Group 3	Group 4	Group 5
Native Enamel (NE) Control	5 s Air-polishing NE_5s_AP	10 s Air-polishing NE_10s_AP	10 s Air-polishing + Rubber polishing NE_RP	10 s Air-polishing + Brush polishing NE_BP
Filtek Ultimate (FU) Control	5 s Air-polishing FU_5s_AP	10 s Air-polishing FU_10s_AP	10 s Air-polishing + Rubber polishing FU_RP	10 s Air-polishing + Brush polishing FU_BP
Enamel Plus HRi (EP) Control	5 s Air-polishing EP_5s_AP	10 s Air-polishing EP_10s_AP	10 s Air-polishing+ Rubber polishing EP_RP	10 s Air-polishing + Brush polishing EP_BP

**Table 3 polymers-14-01643-t003:** Multiple comparisons of mean values of the surface roughness between the different investigated groups of enamel analysed by one-way ANOVA and Tukey’s post-hoc test.

Comparison between Different Treatments	Mean Difference (µm)	*p*-Value	95% Confidence Interval	
			Lower	Upper
Group 1 vs. Group 2	0.09	0.55	−0.09	0.27
Group 1 vs. Group 3	0.35	<0.001	0.17	0.53
Group 1 vs. Group 4	0.41	<0.001	0.23	0.58
Group 1 vs. Group 5	−0.09	0.62	−0.26	0.09
Group 2 vs. Group 3	0.25	<0.01	0.07	0.43
Group 2 vs. Group 4	0.31	<0.001	0.13	0.49
Group 2 vs. Group 5	−0.18	0.05	−0.36	0.001
Group 3 vs. Group 4	0.059	0.86	−0.12	0.24
Group 3 vs. Group 5	−0.43	<0.001	−0.61	−0.25
Group 4 vs. Group 5	−0.49	<0.001	−0.67	−0.31

**Table 4 polymers-14-01643-t004:** Multiple comparisons of mean values of the surface roughness between the different investigated groups of the nanofill Filtek Ultimate analysed by one-way ANOVA and Tukey’s post-hoc test.

Comparison between Different Treatments	Mean Difference (µm)	*p*-Value	95% Confidence Interval	
			Lower	Upper
Group 1 vs. Group 2	−0.28	<0.01	−0.46	−0.09
Group 1 vs. Group 3	−0.46	<0.001	−0.65	−0.28
Group 1 vs. Group 4	−0.003	1.00	−0.19	0.18
Group 1 vs. Group 5	−0.09	0.63	−0.27	0.09
Group 2 vs. Group 3	−0.18	0.05	−0.37	−0.002
Group 2 vs. Group 4	0.28	<0.01	0.09	0.46
Group 2 vs. Group 5	0.19	0.04	0.04	0.38
Group 3 vs. Group 4	0.46	<0.001	0.27	0.65
Group 3 vs. Group 5	0.38	<0.001	0.19	0.56
Group 4 vs. Group 5	−0.08	0.68	−0.27	0.10

**Table 5 polymers-14-01643-t005:** Multiple comparisons of mean values of the surface roughness between the different investigated groups of the microhybrid Enamel Plus HRi analysed by one-way ANOVA and Tukey’s post-hoc test.

Comparison between Different Treatments	Mean Difference (µm)	*p*-Value	95% Confidence Interval	
			Lower	Upper
Group 1 vs. Group 2	−0.93	<0.001	−1.18	−0.67
Group 1 vs. Group 3	−0.96	<0.001	−1.22	−0.70
Group 1 vs. Group 4	0.12	0.64	−0.14	0.38
Group 1 vs. Group 5	−0.95	<0.001	−1.21	−0.69
Group 2 vs. Group 3	−0.03	0.99	−0.29	0.23
Group 2 vs. Group 4	1.05	<0.001	0.79	1.31
Group 2 vs. Group 5	−0.02	0.99	−0.28	0.24
Group 3 vs. Group 4	1.08	<0.001	0.82	1.34
Group 3 vs. Group 5	0.01	<0.001	−0.25	0.27
Group 4 vs. Group 5	−1.07	<0.001	0.81	1.33

**Table 6 polymers-14-01643-t006:** Relative effect size of the factors Material, Treatment, and their interactions on the surface roughness of the investigated resin-based composites analyzed by General Linear Model and Partial Eta-Squared statistics.

Factor	Surface Roughness	
	*p*-Value	Partial ƞ^2^
*Material*	<0.001	0.862
*Treatment*	<0.001	0.885
*Material × Treatment*	<0.001	0.805

## Data Availability

Not applicable.

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
