# Peer review of "Effect of Air-Polishing and Different Post-Polishing Methods on Surface Roughness of Nanofill and Microhybrid Resin Composites"

_polymers, 2022, doi:10.3390/polym14091643_

Round 1

Reviewer 1 Report

The study is innovative and may introduce a valuable information to the current dental literature, however the authors should address the following points:

  • The objectives of the study needs to be re-written for clarity.
  • The authors should add their null hypotheses at the end of the introduction section.
  • The authors didn't mention how third molars were preserved following extraction till testing, since desiccation may affect the validity of the study protocol.
  • How did the authors come up with desired size of the tested specimens (8x5mm and 2mm thickness)? please cite any available standards or similar previous studies.
  • How long the samples were stored prior to testing and what was the used method for storage?
  • Were the samples randomly assigned to the study groups? please elaborate.
  • 5 sample size per each group is quite small, did authors perform pilot study and power analysis.
  • Why did the authors use 5 and 10 seconds air polishing duration for groups 2 and 3. Please cite any available standards or literature that support this protocol.
  • Line 155, the authors stated that 30x30 microns area were checked among the study samples. Were those areas randomly selected?
  • AFM software data in line 162 can be removed since it was mentioned at a previous line.
  • The authors used the untreated groups as controls, were enamel and composite samples in treated groups measured at baseline (before treatment) to detect changes at sample-level?
  • The authors should add a clinical relevance part somewhere in the discussion section.

Author Response

Answers to Reviewer’s comments

Every modification according to the reviewer's kind instructions was indicated with red in the text.

Reviewer#1

  1. The objectives of the study needs to be re-written for clarity. The authors should add their null hypotheses at the end of the introduction section.

According to the reviewer’s kind advice, the objectives were re-written and null-hypotheses were established.

“The purpose of our in vitro study was to test the surface damaging effect of the calcium-carbonate based prophylactic AP method on the surface of nanofill and microhybrid RBCs as well as enamel and to obtain a direct quantitative comparison of the changes in surface roughness using an AFM. Further aim was to investigate the efficiency of one-step and two-step post-air-abrasive polishing processes on surface roughness to re-establish the baseline surface smoothness. The null-hypotheses of our investigation were the followings: (a) there is no difference in the surface roughness of resin composites and native enamel surfaces before and after the air-polishing with calcium-carbonate; (b) there is no difference in the surface roughness among the resin composites with different filler compositions after air-polishing with calcium-carbonate; (c) there is no difference in the surface roughness among the resin composites and on the enamel after different post-polishing procedures.”

  1. The authors didn't mention how third molars were preserved following extraction till testing, since desiccation may affect the validity of the study protocol. How long the samples were stored prior to testing and what was the used method for storage?”

According to the reviewer's comment the storage condition was completed in the text.

“The third molars were immersed to 5.25 % sodium-hypochlorite solution for 5 minutes immediately after the extraction for disinfection, then were stored in 0.9 % sterile physiologic saline solution at 37  ̊C (Cultura Incubator, Ivoclar Vivadent, Schaan, Liechtenstein) for one week right before the slice preparation to avoid desiccation.”

  1. How did the authors come up with desired size of the tested specimens (8x5mm and 2mm thickness)? please cite any available standards or similar previous studies.”

For the AFM and SEM analysis the 8x5x2 mm enamel slices were considered to be optimal, easy to prepare and handle during the examinations, furthermore resistant enough during the manipulation. A relevant reference can support our decision: Rodríguez-Vilchis LE, Contreras-Bulnes R, Olea-Mejía OF, Sánchez-Flores I, Centeno-Pedraza C. Morphological and structural changes on human dental enamel after Er:YAG laser irradiation: AFM, SEM, and EDS evaluation. Photomed. Laser. Surg. 2011, 29, 493-500.

  1. “How long the samples were stored prior to testing and what was the used method for storage?”

The prepared enamel and resin composite samples were stored in distilled water at 37  ̊C (Cultura Incubator, Ivoclar Vivadent, Schaan, Liechtenstein) maximum for one week prior to testing. The text has been supplemented according to the followings:

“All enamel and RBC samples were stored in distilled water at 37ºC for maximum one week prior to testing.”

  1. “Were the samples randomly assigned to the study groups? please elaborate.”

Yes, the samples were randomly divided to the study groups. The manuscript text has been supplemented with this information.

“The prepared samples were randomly divided into five groups (n=5x5).”

  1. “5 sample size per each group is quite small, did authors perform pilot study and power analysis.”

According to a pilot study sample size calculation was used to determine the sample size per each group. The Materials and Methods / Evaluation and statistical analysis section has been supplemented with this information.

“Pilot study results and sample size formula were used to estimate sample size.

Sample size formula:  = 2.5

(z = standard score; α = probability of Type I error = 0.05; z1- α/2 = 1.96; β = probability of Type II error = 0.20; 1−β = the power of the test = 0.80; z1−β = 0.84, M1 = 1.71, s1 = 0.14, M2 = 1.31, s2 = 0.18). By adopting an alpha (α) level of 0.05 and a beta (β) level of 0.20 (power = 80%), the predicted sample size (n) was found to be a total of 2.5 samples per group. Instead of the calculated 2.5 samples, n = 5 per group sample size was selected.”

  1. “Why did the authors use 5 and 10 seconds air polishing duration for groups 2 and 3. Please cite any available standards or literature that support this protocol.”

Although, the NSK company has no exact recommendation for the appropriate application time of air-abrasive on the tooth surfaces, in their oral hygiene therapy guide 6,5 sec effective    average time period for biofilm removal was reported [https://www.nskdental.com/admin/wpcontent/uploads/Category_Catalog_Oral_Hygiene.pdf]. According to this information we selected the 5 and 10 s air-polishing time to detect the differences of a shorter and longer exposure time. This decision was also supported by a corresponding literature: Bühler J, Schmidli F, Weiger R, Weiger C. Analysis of the effects of air polishing powders containing sodium bicarbonate and glycine on human teeth. Clin. Oral. Investig. 2014, 19, 877-85.

  1. “Line 155, the authors stated that 30x30 microns area were checked among the study samples. Were those areas randomly selected?”

Yes, the examined areas were randomly selected. According to the reviewer’s remark, the text was completed with this information.

“For each sample, 30 μm x 30 μm images were scanned in three randomly selected areas at a resolution of 512 x 512 pixels.”

  1. “AFM software data in line 162 can be removed since it was mentioned at a previous line.”

Thank you for this remark, the redundant data has been removed from the text.

  1. “The authors used the untreated groups as controls, were enamel and composite samples in treated groups measured at baseline (before treatment) to detect changes at sample-level?”

Although, the reviewer’s suggestion is absolutely relevant, during our study design the control groups were selected as baseline for comparison to the treated groups. The reason of our decision is the negligible difference of repeated measurements within the control groups regarding the surface roughness (S.D. ~ 0.03-0.06).

  1. “The authors should add a clinical relevance part somewhere in the discussion section.”

According to the reviewer’s kind advice, the Conclusion section has been supplemented with a clinical relevance part:

“As a clinical relevance it can be concluded, that air-abrasion can compromise the surface smoothness of enamel and RBC restorations, however, post-polishing with two-step rubber polishers can re-establish the surface smoothness which may avoid increased plaque accumulation and discoloration.”

Reviewer 2 Report

Dear Authors.

Congratulations on your work which, I found interesting.

Manuscript: Effect of air-polishing and different post-polishing methods on surface roughness of nanofill and microhybrid resin composites, it is well written with an adequate structure as a scientific paper demands.

I have some minor revisions to propose to you to improve your work. Please refer to the following comments:

  • Please explain why the enamel and composite samples had different dimensions.
  • Was the power of the LED lamp used for polymerization tested? The range of 850-1000 mW/cm2 is quite large and could have an impact on the final results.
  • In group 4, polishing with fine and extra fine rubber lasted 10 seconds, or every rubber 10 seconds, so a total of 20 seconds?
  • Why were the enamel and composite samples evaluated under different magnifications (x1000 and x2000)?
  • The literature is old, with many citations of works from before 2012 - please consider updating it.

Author Response

Answers to Reviewers’ comments

Every modification according to the reviewer's kind instructions was indicated with red in the text.

Reviewer#2

  1. “Please explain why the enamel and composite samples had different dimensions.”

Thank you for the observation. The examined area of each sample by AFM was 30x30µm, thus both the enamel and composite samples provided enough surface for this analysis. The only reason of the bigger sized enamel samples was the easy of handling during enamel slice preparation from the extracted third molars.

  1. “Was the power of the LED lamp used for polymerization tested? The range of 850-1000 mW/cm2 is quite large and could have an impact on the final results.”

Thank you for this important question. Yes, the irradiance of the LED source was monitored before and after curing with a radiometer (Cure Rite, Dentsply, Milford, DE, USA). The indicated range is given by the manufacturer, however, the actual irradiance measured by the radiometer was 1050±10 mW/cm2. The text has been corrected with the actually measured irradiance instead the range given by the manufacturer.

“The irradiance of the LED unit was 1050±10 mW/cm2 and monitored before and after curing with a radiometer (Cure Rite, Dentsply, Milford, DE, USA).”

  1. “In group 4, polishing with fine and extra fine rubber lasted 10 seconds, or every rubber 10 seconds, so a total of 20 seconds?”

Thank you for the question, the total polishing time period lasted for 20 seconds, each rubber polishers were used for 10 seconds. The text has been modified to clarify this information.

“In Group 4 a two-step rubber diamond polisher (fine – 10 seconds, 8-32 µm grit size, Kenda Nobilis, Kenda AG, Vaduz, Liechtenstein; then extra fine – 10 seconds, 4-8 µm grit size, Kenda Unicus, Kenda AG, Vaduz, Liechtenstein) was used. In Group 5 a one-step polisher, an abrasive-impregnated polishing brush with in-build silicone carbide abrasive particles (Occlubrush cup, KerrHawe SA, Bioggio, Switzerland) was used for 10 seconds to polish the surface.”

4.“Why were the enamel and composite samples evaluated under different magnifications (x1000 and x2000)?”

The surface of the enamel samples was analyzed under 1000x magnification to be able to visualize bigger areas of the affected enamel prisms and see an overview picture of the changes in the surface structure. The composite samples showed structural changes in the microsturctural level. The changes of the resin composite (disintegrated resin matrix and the filler particles became visible at the surface) were more detectable and spectacular at 2000x magnification.

5.“The literature is old, with many citations of works from before 2012 - please consider updating it.”

According to the reviewer’s kind advice, some of the older references were changed to a newer one.

4.Giti R, Dabiri S, Motamedifar M, Derafshi R. Surface roughness, plaque accumulation, and cytotoxicity of provisional restorative materials fabricated by different methods. Plos One 2021, 16, e0249551.

19.Yang KI, Park DY, Kim BO, Yu SJ. Clinical and microbiological study about efficacy of air-polishing and scalingand root-planing. Int. J. Oral. Biol. 2015, 40, 93-101.

24.Sahrmann P, Ronay V, Schmidlin PR, Attin T, Paqué F. Three-dimensional defect evaluation of air polishing on extracted human roots. J Periodontol. 2014, 85, 1107-14.

39. Deleted
